# VALUE-AWARE TRANSFORMERS FOR 1.5D DATA

## ABSTRACT

Sparse sequential highly-multivariate data of the form characteristic of hospital in-patient investigation and treatment poses a considerable challenge for representation learning. Such data is neither faithfully reducible to 1D nor dense enough to constitute multivariate series. Conventional models compromise their data by requiring these forms at the point of input. Building on contemporary sequence-modelling architectures, we design a *value-aware transformer*, prompting a reconceptualisation of our data as *1.5−dimensional*: a token-value form both respecting its sequential nature and augmenting it with a quantifier. Experiments focused on sequential in-patient laboratory data up to 48hrs after hospital admission show that the value-aware transformer performs favourably versus competitive baselines on in-hospital mortality and length-of-stay prediction within the MIMIC-III dataset.

## 1 INTRODUCTION

Modelling the sequence of events in electronic health records (EHRs) poses fundamental challenges to contemporary sequence models owing to the complex, heterogeneous nature of the data. Even choosing a good input representation for such data presents difficult trade-offs. The variety of data shapes encountered within the records for different patients and across different sections of their visit is substantial. While there are some variables, such as patient demographics, that one might reasonably expect all patients to have an entry for, the heterogeneity of structure sets in as soon as the patient's individual health status is more intricately explored. During an admission to hospital, a series of laboratory tests (hereafter referred to as 'labs' for succinctness) may be conducted on a patient: the number of possible types of test might be large (in the hundreds), these may be measured repeatedly, once, or not at all. Hospital stays may vary significantly in length and this variation is not necessarily indicative of severity. Very ill patients may have short survival, or may be artificially sustained on life support in expectation of a recovery that fails to arrive. There are further complications: some tests may have been conducted in advance - attributed to their historical health record; or recorded in a different modality of their health record (e.g. notes, charts, etc.). The vast complexity and heterogeneity of EHR data presents representation learning with fertile ground for innovation.

### 1.1 OUR CONTRIBUTIONS

This paper seeks to address a particular data regime which is under-studied: that of sparse sequential and highly-multivariate data, where the total number of variables typically outstrips the total number of measurements for a given data instance. Table 6 illustrates this for the MIMIC-III lab-events dataset, where the mean number of completely unmeasured variables for a given visit is 697 out of a possible 753. As we shall describe, conventional approaches to modelling this data make severe simplifying assumptions, or rely heavily on imputation. Many contemporary works restrict both the variables selected and the population studied, so as to reduce missingness. To address these issues, we:

- Present a data encoding that effectively captures sparse sequential highly-multivariate data while avoiding considerable imputation.

- Propose neat adaptations to make a standard transformer architecture value-aware, equipping it to learn and generate this data encoding.

- Provide an empirical study on a real-world dataset demonstrating that a pre-trained value-aware transformer performs competitively on challenging downstream predictive tasks.

## 1.2 RELATED WORK

This work sits on the intersection of a number of exciting areas of research: representation learning for healthcare, transformer architectures, sparse sequence modelling. Here we try to do justice to some of the many relevant works.

**Representation learning for EHRs.** Researchers seeking to make secondary use of the signal within EHRs have paid particular attention to the developments in deep learning (Shickel et al., 2017; Si et al., 2020), with applications ranging from unsupervised patient stratification Landi et al. (2020) to learning personalised comorbidity networks (Qian et al., 2020). The wave of excitement set in motion by transformer architectures (Vaswani et al., 2017) quickly propagated to the healthcare world. Large language models such as BERT (Devlin et al., 2018) have been subjected to rounds of secondary pre-training on medical corpora (Alsentzer et al., 2019), with the aim of harnessing signal locked up in the idiosyncratic natural language formats such as clinical notes. By encoding events as tokens, transformers can also be utilised for studying EHR event sequences. When arranged to constitute a sentence, these tokens capture the pattern of a patient's stay. This approach can be pursued at various levels of granularity, for instance Li et al. (2020) use BERT to model sequences of diagnoses over multiple visits. Applying the same methodology at a finer grain presents a natural model for within-visit medical event sequences such as labs. But while knowledge of the performance of a lab test conveys information, it is the lab test *together with its value* that is the more complete indicator of the health status of a patient. Choi et al. (2020) made progress on integrating labs and their values using a graph convolutional transformer, modelling the graphical and hierarchical structure of EHRs in an exciting piece of work. However, their method was not designed to model repeat measurements of labs - a common phenomenon within the natural course of medical investigation and treatment, motivating our work.

**Healthcare benchmarks.** In order to objectively compare systems such as those described above, EHR-specific benchmark tasks have been proposed, many based on the MIMIC dataset Johnson et al. (2016). Of particular relevance here are the prediction tasks for in-hospital mortality and length-of-stay proposed in Harutyunyan et al. (2019); Purushotham et al. (2018). In related work, and with a view to further increasing reproducibility, Wang et al. (2020) provided a pre-processing pipeline that addresses concerns such as unit conversion, outlier filtering and aggregation that are common in these tasks. The considerable heterogeneity of structure in EHRs means that the benchmarking papers need to make many design and preprocessing choices for the cohorts selected, the variables used, and the definitions of prediction targets. Not all of these choices are desirable for applications: e.g. in Harutyunyan et al. (2019) labels are defined in such a way that the events of interest could occur within the data visible to the model rather than being restricted to predicting future events, while in Purushotham et al. (2018) their largest feature set includes only 11 of the approx. 700 raw features available in the MIMIC-III lab tests, chosen due to their minimal missingness rate. Some of these considerations (targets labels) we would like to refine so as to provide more utility, others (forced variable selection and aggregation) do not align with the purpose of this work, which is to remove the need for cherry-picking of variables. This makes direct comparison with these works challenging - a point which other authors have highlighted, even when using largely similar cohorts, variables and identical prediction targets (Che et al., 2018).

**Irregular times series.** A closely related problem is that of studying irregularly-sampled multivariate time-series. Shukla & Marlin (2020) detailed three different types of input representations: series-based, vector-based and set-based, capturing time, variable and value for a range of scenarios. Horn et al. (2020) focus on the set-based approach considering each times series to be a bag-of-events, proposing a model which applies a set function together with a single attention mechanism between elements of the set, topped with a classifier. In their experiments, an out-of-the-box transformer is state-of-the-art for the Physionet 2012 prediction task (Goldberger et al., 2000), and performs competitively on a MIMIC-III mortality prediction task (based on Harutyunyan et al. (2019)). Another approach is elicited in (Che et al., 2018), where they introduce a GRU-based architecture, designed with inbuilt decay mechanisms. Their method is attuned to time-series variables such as flow data or regularly-sampled labs and relies on the imputation of missing values, which, for our data regime would constitute on average 98% of the input (see Table 6).

## 2 DATA GREATER THAN 1D

Sequences of words such as those found in natural language are neatly represented as 1D data using tokens. Sequences of lab results such as those found early in EHRs are neatly represented as (token, value) pairs, which might suggest that the ideal representation of this data should a multivariate series with one dimension per lab type. To model it in this form would require considerable imputation due to the data's sparsity. However, the presence or missingness of particular labs in a sequence are informative: they are requested by healthcare professionals guided by clinical intuition that the EHR may not capture. By imputing the data, this valuable signal is weakened. Perhaps we can make do with just a 1D form - and if so, how should that 1D be constituted? These are the key questions that a practitioner hoping to model in our data regime faces. As we will discuss in this section, when the data is sparse and highly-multivariate, the choice is not clear *a priori*. For the motivational problem of lab-result sequences there may be only few or a large number of measurements with multiple repeat samples of the same lab - characteristics which may contain important information.

We close the section by proposing a variation on the conventional approaches, which we refer to as "1.5−dimensional" data: a form preserving the sequential nature of the lab orders, together with information about the values, but without the need for full multivariate or sparse vector-based sequence representations and the imputation issues that would ensue.

### 2.1 CHARACTERISTICS OF LAB-RESULT SEQUENCES

In-patient lab-result sequences are generated by a process of hospital investigation and treatment which imbue them with particular characteristics. Relevance: the labs are pertinent to the health investigation being conducted on the patient. Only a small proportion of the all of the labs available might be obtained during any given visit. Repeats: these relevant measurements may be taken once, or repeated many times. Pragmatism: difficult or inconvenient labs will not be taken unless absolutely necessarily. This leads to a rare-token problem also commonly encountered in natural language. If we believe that these characteristics deliver valuable information, then the way that we represent our data should not compromise them.

### 2.2 COLLAPSING DATA DOWN TO 1D

A common way to deal with this difficult data dimensionality is to 'collapse' it to 1D. There are two natural ways of doing this: the horizontal collapse and the vertical collapse, illustrated in Figure 1.

**Horizontal collapse.** The horizontal collapse retains the sequential structure of the data, and removes the value aspect. One form of the horizontal collapse is to ignore the value from the token-value pair. We then have sequences of lab orders, and the premise is that there is sufficient information contained in the *presence* of a lab order to inform the underlying state of a patient. This capi-

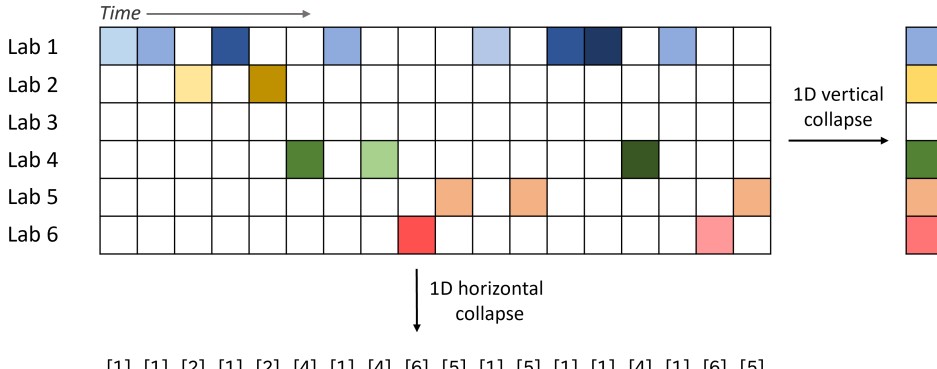

Figure 1: The 1D horizontal data collapse forms a sequence of lab tokens while ignoring their values; a 1D vertical collapse takes a summary statistic (e.g. the mean) for each lab sequence.

talises on recent advancements in NLP and has formed the basis for some of the medical sequence-modelling applications mentioned in Section 1.2.

**Vertical collapse.** An alternative form of 1D collapse is to collate repeated values of a variable using a summary measure (such as the mean, median, max, most-recent) and then use these summary measures – one per variable – as the input to a model. This representation respects the unique value scales on a per-variable basis, however it compromises the sequential nature of the data and collapses repeated measurements.

### 2.3 IMPUTING FOR MULTIVARIATE SEQUENCES

An enticing candidate for the representation of *(token, value)* pairs is as a multivariate sequence with one sequence dimension per token. This retains each variable's independent value-scale and the innate temporally-ordered nature of the data. However, the sparsity of our specific data regime is problematic, affecting its suitability: a program of severe imputation is required in order to use this input form. This commonly takes the form of sequential forward- or backward-filling of data; per-variable mean imputation; or the use of sentinel values, the choice of which will necessarily depend on the variable in question (so as not to inappropriately assign a sentinel value that is commonly attained by *true* data points). Imputation is not necessarily evil, but does require careful application, even in a relatively dense data regime. Not all missingness is made equal and even the most advanced general-purpose imputation schemes rely on assumptions (MCAR or MAR: see Mohan & Pearl (2021)) which are often unrealistic in a healthcare setting. Unrealistic imputation can have unexpected downstream implications (Caruana, 2021) which can be dangerous for critical applications. Even a sparse vector-based multivariate representation such as that detailed in Shukla & Marlin (2020) would require on average 98% of its input to be imputed for our data (see Table 6).

### 2.4 1.5D DATA

Both the 1D collapse and the multivariate sequence approaches introduce their own issues or simplifications for sparse sequential highly-multivariate data. We propose to interpolate these data regimes and use an alternative data format: ([TOK], [VAL]), where [TOK] is the name of the variable and where [VAL] is its value, which has been transformed and discretised in such a way that it is meaningfully comparable across all variables. Such an approach:

- retains the sequential nature of the data: it is a 1D sequence of tokens.
- removes the need for considerable imputation (cf. Tables 1 and 6)
- reduces the heterogeneous value dimension to a common scale, contextualised by its token.

For numeric labs, the value tokenisation can be achieved via *quantilisation*: conditional population statistics for the lab results can be used to attribute a category to each of the values, depending on which quantile the value lies in. If there are no prior distributional statistics available, they can be extracted from the training set data (for simplicity and generality we took this approach in our analysis). For categorical labs we can encode them in any manner which is consistent. Where no category can be associated from those mentioned previously, a sentinel category can be used - this may be necessary in the case where a result is not recorded, or the lab itself is not seen within the training set.

Table 1: Comparison of number of inputs required for 1D, 1.5D and regularly-sampled multivariate sequence representations, where: $l$ is # variables, $n$ is number of sequence steps of the model, $m$ is # data entries in a particular sequence. See Table 6 for an analysis of our dataset.

| Data representation | # Inputs | # Imputations | Signal compromised |
|---|---|---|---|
| 1D (vertical) | $l$ | $\in [l-m, l-1]$ | sequence structure & repeat measures |
| 1D (horizontal) | $n$ | $n-m$ | values of measurements |
| 1.5D | $2n$ | $2(n-m)$ | raw value-scales |
| Multivariate sequences | $l \cdot n$ | $l \cdot n - m$ | investigative pattern |

The discrete categories chosen should be chosen dependent on the application. Since our variables are physiological measurements we choose to use a quantilisation that accentuates the extremities of the distributions, assigning the following discretisation:

- `[XLOW]`: the value lies below the 10% percentile
- `[LOW]` : the value lies between the 10% and 25% percentiles
- `[MID]` : the value lies between the 25% and 75% percentiles
- `[HIGH]` : the value lies between the 75% and 90% percentiles
- `[XHIGH]` : the values lies above the 90% percentile.

The token `[UNK]` was assigned if the lab did not feature in the train set, or if there was no numeric lab value associated with the lab. This specific quantilisation should not be read as definitive: other natural candidates are using quartiles, deciles, or a normal/abnormal discretisation, and should be informed by the modelling application.

Multiple events might approximately coincide: to break these ties we used a consistent ordering of labs. For our application the order of coincident tests should have no direct bearing on the informativeness of the tests themselves, but a random ordering would make it harder for the model to learn these patterns. This can be more explicitly dealt with using temporal embeddings (see Appendix D).

The `[VAL]` tokens should contextualised by their associated `[TOK]`. The sequence of `[VAL]`s, while plausibly carrying some autoregressive signal, are not in-of-themselves readily interpretable as the sequential data of a single variable. They augment a 1D sequence of tokens with values, hence the apt moniker "1.5D data".

## 3 VALUE-AWARE TRANSFORMERS

Transformer architectures in their various guises have shown exceptional performance on tokenised sequence modelling tasks, but they lack a principled approach for learning *(token, value)* sequences. The 1.5D data form lays a foundation for doing so. We outline guiding principles which provide a basis for utilising this input form based on the discussion in Section 2.4, then design a neat and minimalistic transformer architecture for learning and generating it.

### 3.1 DESIGN PRINCIPLES

1. Tokens are primary. First, there were tokens, and then came their values.

Principle 1 makes a statement about precedence. The sequence of values `[VAL]`, by themselves, are bereft of meaning and without a clear autoregressive interpretation. To benefit from these qualities the model must contextualise them using their respective tokens.

2. Values should be loosely-coupled to their token.

Loose-coupling is most easily defined by contrast with a *strong-coupling* where each token `[TOK]` and its quantifier, the value `[VAL]`, is explicitly concatenated during preprocessing to form a new token `[TOK-VAL]`. Strong-coupling drops a fundamental symmetry present in the raw data: the information that the pairs (`[TOK]`, `[VAL1]`) and (`[TOK]`, `[VAL2]`) are of the same variable is not provided to the model if using this representation. This concatenation approach scales poorly with the number of distinct tokens and values tokens[1]. At the other extreme there is *no-coupling*. No-coupling is to treat the tokens and values as completely separate entities. Modelling them in this way both causes the 1.5D form to lose its meaning.

3. Values should have the opportunity to influence the autoregressive learning of the token sequence.

While the core autoregressive properties of the 1.5D sequence lie amongst the tokens, there may be useful information in their associated values which we should aim to capitalise on.

---

[1] For further comparison with strong-coupling and empirical experiments see Appendix E.

### 3.2 Tailoring a value-aware transformer

The temporal nature of our sequence data make a decoder-only style transformer a compelling candidate architecture. Large decoder-only models such as GPT-3 (Brown et al., 2020) have demonstrated astonishing performance, especially for natural language generation, and while data generation is not the primary use-case here, the future-masking inductive bias is an appropriate one for event data. We highlight three core components that adapt a plain decoder-only transformer to the 1.5D data:

**Embeddings.** To respect principle 1 and ensure loose-coupling of the (`[TOK]`, `[VAL]`) pairs we use separate embedding layers for the tokens and the values. These two embeddings are then concatenated. Only after this concatenation is a positional embedding added, so that both parts have their position consistently encoded.

For our experiments the number of tokens is $\approx 700$ and the number of values $\approx 10$ - we map the tokens to $\mathbb{R}^{100}$ and the values to $\mathbb{R}^5$ to force some compression. Deciding the optimal dimensionality of embedding is an open problem, but many embedding algorithms do not suffer much loss of performance due to over-parameterisation Yin & Shen (2018).

**Attention.** The loosely-coupled embedding concatenation means that information about the values is included within the embeddings that are propagated through the attention layers. This is key so that principle 3 is respected. We use a now established bag of tricks (Wang, 2021) with the normalise-attend-feedforward (with residuals) structure for the attention layers constituting our value-aware transformer.

**Output layers.** The choice of autoregressive prediction head for the transformer is another crucial design consideration. Both values and tokens should be predicted, and how this is done has a significant effect on the overall learning of the model because of its proximity to the loss calculation.

Splitting the final attention layer's output embedding into two parts proved to work better than sharing the entire embedding by some margin. In respect of principle 1, the lion's share is dedicated to next-token prediction and the remaining part of the embedding used for next-value prediction. Dedicated token-prediction and value-prediction layers are then attached to their respective segment of the embedding. To respect the priority of tokens, the value-prediction layer is conditional on the predicted token, explicitly including it whether correct or incorrect.

Finally, we control the relative weight given to the cross-entropy losses for the tokens $\mathcal{L}_{[\text{TOK}]}$ and the values $\mathcal{L}_{[\text{VAL}]}$ using a hyperparameter $\gamma \in [0, 1]$:

$$\mathcal{L} = \gamma \mathcal{L}_{[\text{TOK}]} + (1 - \gamma)\mathcal{L}_{[\text{VAL}]}. \tag{1}$$

#### 3.2.1 Pre-training

With a view to later fine-tuning tasks we include one type of special token: an `[EOS]` token for the end-of-sequence. The `[EOS]` has a distinguished role as the token upon whose output embedding we perform classification or regression. There is a different `[EOS]` for the tokens ($\in \mathbb{R}^{100}$) and for the values ($\in \mathbb{R}^5$).[2]. The last special token we include is a `[PAD]` - to fill up empty slots in the sequence. The `[PAD]` class is not used when calculating the cross-entropy losses $\mathcal{L}_{[\text{TOK}]}$ and $\mathcal{L}_{[\text{VAL}]}$. During pre-training the model seeks to minimise the total loss, equation 1.

#### 3.2.2 Fine-tuning

Fine-tuning a value-aware transformer for a classification task consists of replacing the autoregressive output layers with a classification head, directly harnessing the embeddings output from the top attention layer. There is a small but important difference in approach from common bidirectional language models. There, the first token is often denoted as `[CLS]` and is imbued with a special classification function, with a classifier head being attached to it (Devlin et al., 2018). Because we are using a (future-masked) decoder-style transformer, classification on a prepended start-of-sequence token would be useless: during fine-tuning, the model would see exactly the same input for each data instance, hence is unable to learn anything. In our setting classification is instead done on the

---

[2]These are always concatenated together during the input embedding stage.

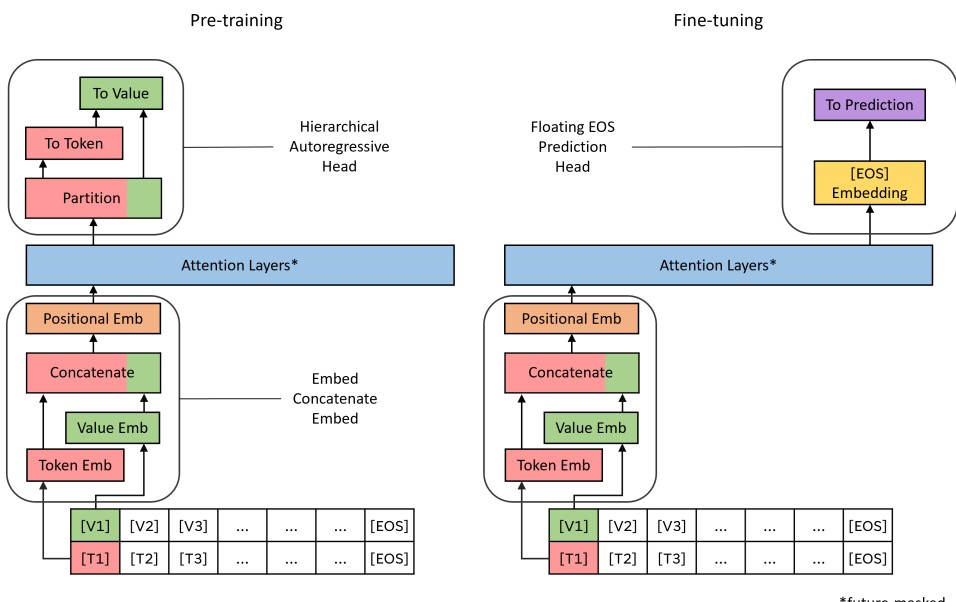

Figure 2: Architecture diagrams for pre-training and fine-tuning a value-aware transformer

[EOS] token: we use a floating classification head which attaches itself at this position, taking as its input the output embedding of [EOS]. We use a simple but flexible one-hidden-layer feed-forward neural net with an output layer to the number of classes relevant for the classification problem. For regression tasks we use exactly the same set up except with a single class output and a further non-linearity. For fine-tuning the model seeks to minimise a problem-dependent loss. For our experiments we used a propensity-weighted cross-entropy loss for imbalanced classification problems and a MSE loss for the regression problem.

## 4 EXPERIMENTS

MIMIC-III ('Medical Information Mart for Intensive Care')[3] is a core resource for machine-learning researchers pursuing the advancement of healthcare. It is a large publicly accessible anonymised database comprising comprehensive information relating to patients admitted to the critical care unit at a large US hospital. The heterogeneity of data structure encountered within it are exemplar of the electronic health records of healthcare systems across the world.

### 4.1 MIMIC-III LAB-EVENTS

Lab-event sequences occurring early within an admission to hospital embody the sparse sequential highly-multivariate data characteristics described above, making the MIMIC-III lab-events an excellent dataset to benchmark on. We focus our experiments on the lab-events data which is available up until 48 hours after admission, pre-training a value-aware transformer to learn its autoregressive structure. We then tackle three challenging downstream problems by fine-tuning the value-aware transformer. The first and second are classification tasks: in-hospital mortality prediction, both death within 3 days, and death after 7 days; the third is a regression task: predicting the remaining length-of-stay (LOS) at 48 hours.

### 4.2 EXPERIMENT 1: PRE-TRAINING

In the first experiment we investigated whether incorporating values assists in the prediction of tokens by comparing the value-aware transformer to a plain transformer which uses the same base

---

[3]Available at https://physionet.org/content/mimiciii/1.4

Table 2: Pre-training performance: value-aware transformer vs. a transformer on Dev. set. $\mathcal{L}_{\texttt{[TOK]}}$ indicates the token cross-entropy loss, and $p_{\texttt{[TOK]}}$ the corresponding average probability of correct next-token prediction.

| Model | Data dim. | $\mathcal{L}_{\texttt{[TOK]}}$ | $p_{\texttt{[TOK]}}$ | $\mathcal{L}_{\texttt{[VAL]}}$ | $p_{\texttt{[VAL]}}$ |
|---|---|---|---|---|---|
| Random | | – | 0.0013 | – | 0.1250 |
| Transformer (depth 4) | 1 h | 1.331 | 0.2642 | – | – |
| VA-transformer (depth 4) | 1.5 | 1.322 | 0.2674 | 1.347 | 0.2600 |
| Transformer (depth 6) | 1 h | 1.319 | 0.2666 | – | – |
| VA-transformer (depth 6) | 1.5 | **1.291** | **0.2750** | **1.332** | **0.2639** |

architecture. We captured their respective token cross-entropy losses $\mathcal{L}_{\texttt{[TOK]}}$ for comparison. We made the competition as fair as possible, choosing the same fixed input sequence length and their respective best learning rate hyperparameters with all other settings kept the same. We let each run until a fixed early stopping threshold on the Dev. set was met. Further details may be found in Appendix B.

The results in Table 2 demonstrate that the value-aware transformer provides a promising improvement over the transformer, with better average next token prediction for both the small (depth 4) and large (depth 6) versions of the model. Comparing the two depth 6 models, the value-aware transformer provides a 3.2% boost over its transformer sibling in the average probability of correct token prediction, whilst also generating values with considerably greater than random probability.

The successful outcome of Experiment 1 demonstrates that there is both valuable signal in the 1.5D representation, and that the value-aware transformer is able to exploit it. The second and third experiments test whether we can channel its superior generative modelling into a performance gain on downstream tasks.

### 4.3 EXPERIMENT 2: IN-HOSPITAL MORTALITY PREDICTION

In-hospital mortality is a challenging clinical outcome to predict because of its highly-imbalanced nature: thankfully a minority of the patients admitted to hospital die. Just 0.84% of visits in our Train set result in death within 3 days of admission. Accuracy is a poor metric of success here - we can attain 99.16% by always predicting that there is no death. Instead we optimise for balanced accuracy, using a Train set propensity-weighted binary cross-entropy loss. We consider two targets: mortality $\leq 3$ days (which has Train set propensity of 0.0084), and mortality $> 7$ days (whose Train set propensity is 0.0496).

We compare the performance of a fixed-specification classifier (a one-hidden layer feed-forward neural network with hidden dimension 100) when trained on various types of 1D vertical collapse (listed as 'FFNN'), with the use of the same classifier head for the fine-tuning of the plain transformer model (using the 1D horizontal representation) and the value-aware transformer (using the 1.5D representation).

The value-aware transformers clearly outperform the other methods for these two tasks, as shown in Table 3. Interestingly, the FFNNs using the quantilised values perform better than using when using the raw values, likely due to a mixture of measurement sparsity and a sensitivity to value extremities, indicating that there is some merit to the discretisation scheme we chose in Section 2.4 for this application.

### 4.4 EXPERIMENT 3: LOS PREDICTION

To round off our experiments we choose a regression task, and length-of-stay is another core problem faced by caregivers. The accurate prediction of LOS enables more efficient delivery of care. When the hospital is under strain or run inefficiently there is a negative impact on patient outcomes.

We use the same fixed-specification regression network for all models. This has the same architecture as for experiment 2, the only differences being a single-class output and a final soft-plus

Table 3: In-hospital mortality classification performance on the Test set, with all metrics multiplied by 100 for ease of reading and the mean $\pm$ std reported over 5 runs.

| Model | Data | Mortality $\leq$ 3 days | | Mortality $>$ 7 days | |
|---|---|---|---|---|---|
| | | Bal-acc | AUROC | Bal-acc | AUROC |
| FFNN raw values (mean) | 1 v | $57.7 \pm 4.4$ | $61.0 \pm 3.3$ | $62.4 \pm 3.0$ | $66.1 \pm 5.4$ |
| FFNN quants (mean, one-hot) | 1 v | $54.6 \pm 6.4$ | $73.6 \pm 1.4$ | $70.4 \pm 0.5$ | $77.9 \pm 1.5$ |
| FFNN quants (mean, integer) | 1 v | $60.3 \pm 2.8$ | $70.7 \pm 3.0$ | $68.6 \pm 0.7$ | $77.2 \pm 0.6$ |
| Transformer (depth 4) | 1 h | $66.6 \pm 1.3$ | $76.1 \pm 0.7$ | $69.5 \pm 0.6$ | $77.7 \pm 0.2$ |
| Transformer (depth 6) | 1 h | $66.3 \pm 2.0$ | $75.2 \pm 1.3$ | $69.9 \pm 1.0$ | $77.6 \pm 0.6$ |
| VA-Transformer (depth 4) | 1.5 | $\mathbf{72.5 \pm 3.9}$ | $\mathbf{82.5 \pm 2.0}$ | $\mathbf{72.3 \pm 1.6}$ | $\mathbf{80.8 \pm 0.3}$ |
| VA-Transformer (depth 6) | 1.5 | $\mathbf{71.6 \pm 2.5}$ | $\mathbf{81.3 \pm 1.2}$ | $\mathbf{72.4 \pm 0.5}$ | $\mathbf{81.0 \pm 0.3}$ |

Table 4: Remaining length-of-stay regression performance on Test set.

| Model | Data dim. | Remaining LOS at 48hrs | |
|---|---|---|---|
| | | MSE | $R^2$ |
| FFNN raw values (mean) | 1 v | $158.6 \pm 6.6$ | $0.091 \pm 0.038$ |
| FFNN quants (mean, one-hot) | 1 v | $131.5 \pm 0.9$ | $0.246 \pm 0.005$ |
| FFNN quants (mean, integer) | 1 v | $131.6 \pm 1.3$ | $0.246 \pm 0.007$ |
| Transformer (depth 4) | 1 h | $132.4 \pm 1.7$ | $0.241 \pm 0.010$ |
| Transformer (depth 6) | 1 h | $131.5 \pm 1.2$ | $0.246 \pm 0.007$ |
| VA-Transformer (depth 4) | 1.5 | $\mathbf{126.9 \pm 1.4}$ | $\mathbf{0.273 \pm 0.008}$ |
| VA-Transformer (depth 6) | 1.5 | $\mathbf{125.2 \pm 3.0}$ | $\mathbf{0.282 \pm 0.017}$ |

non-linearity, for the reason that remaining-LOS is a non-negative real number. We compare this regression network's performance when the input takes the form of the 1D vertically collapsed data with a pre-trained transformer trained on the 1D horizontally collapsed data and a value-aware transformer trained on the 1.5D representation.

As can be seen from Table 4, both the depth 4 and the depth 6 value-aware transformer surpass the competitor models by a healthy margin on the LOS regression task, providing further compelling evidence that additional information of worth is retained in the 1.5D input representation and that the value-aware transformer is able to convert it into superior downstream performance across a range of tasks.

## 5    CONCLUSION

This paper is devoted to sparse sequential highly-multivariate data - a form which is found throughout the healthcare domain and whose value is difficult to unlock. We have analysed conventional approaches to modelling this data using representation learning methods, highlighting the 1D vertical and horizontal collapses and the considerable imputation necessitated by models using a multivariate sequence input.

Offering a fresh perspective, we have reconceptualised the raw data as 1.5-dimensional and presented a token-value form which retains both sequential structure and information about the values, without forcing considerable imputation before the point of input to a model. We have outlined principles for the appropriate modelling of this data form, and based on these principles, made targeted adaptations to a standard transformer architecture, efficiently tailoring it to the generation of 1.5D data.

In a series of experiments on the MIMIC-III lab-events dataset we have demonstrated both the appropriateness of the 1.5D data representation and the suitability of the *value-aware transformer* by clearly surpassing competitive baselines both as an autoregressive model, and on the challenging real-world in-hospital mortality (classification) and length-of-stay (regression) prediction tasks.

REPRODUCIBILITY STATEMENT

The codebase supporting this work has been made available with the submission. Included within it are: the scripts used to pre-process the data; a va_transformers package which is based on the great work of Wang (2021); the scripts used for the pre-training, fine-tuning and baseline experiments. Supplementary details of the pre-processing and experimental settings are outlined in Appendices A and B. The MIMIC-III dataset available at `https://physionet.org/content/mimiciii/1.4/` after suitable completion of training.

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

## A   SUPPLEMENTARY DATASET INFORMATION

For a visit to be included in our dataset it must have lasted at least 48 hours from admission to discharge or death. It must have at least ten lab-events recorded and death must not occur at 48 hours or before. Those visits that met these criteria were then randomly split on a patient identifier into Train, Dev. and Test sets with the proportions 0.8 / 0.1 / 0.1. Some patients have multiple visits so it was important to partition based on the patient so as not to introduce a data leak. The partition statistics are detailed in Table 5. For our experiments, minimal processing of the data was performed so as not to collapse exploitable signal within the raw data (see Section B for further details). We use 753 raw lab-event variables, and do not supplement them with static variables such as demographics, since this study is focused on the effective utilisation of sequential data.

Table 5: MIMIC-III labs data statistics after preprocessing

| Partition | Train | Dev. | Test |
|---|---|---|---|
| # Patients | 32989 | 4124 | 4124 |
| # Visits | 41403 | 5164 | 5072 |
| # Labs/visit mean | 142.8 | 142.3 | 143.8 |
| # Labs/visit median | 127.0 | 128.0 | 129.0 |
| # Labs/visit 90% | 248.0 | 245.7 | 250.0 |
| Remaining LOS mean (hrs) | 219.3 | 227.7 | 226.8 |
| Remaining LOS median (hrs) | 125.6 | 127.3 | 130.4 |
| Mortality $\leq$ 3 days | 0.00841 | 0.00833 | 0.00848 |
| Mortality $>$ 7 days | 0.04959 | 0.04493 | 0.05205 |

Table 6: Comparison of the number of imputations required for 1D, 1.5D and multivariate series data representations for the MIMIC-III lab-events data. where: $l = 753$ is # variables, $n = 250$ is number of sequential steps (cf. Table 1). Vector-based is detailed in Shukla & Marlin (2020) and has a varying sized input - we report the mean.

| Data representation | # Inputs | # Imputations required/visit (Train set) | | | |
|---|---|---|---|---|---|
| | | Mean | (% Input) | Median | (% Input) |
| 1D  (vertical) | 753 | 696.9 | (92.6%) | 699 | (92.8%) |
| 1D  (horizontal) | 250 | 115.4 | (46.1%) | 123 | (49.2%) |
| 1.5D | 500 | 230.7 | (46.1%) | 246 | (49.2%) |
| Multivariate sequences | 188250 | 188107.2 | (99.9%) | 188123 | (99.9%) |
| Vector-based | 8020.7 | 7877.9 | (98.0%) | 6627 | (98.1%) |

## B   EXPERIMENTAL SETTINGS & HYPERPARAMETERS

**Finetuning for the value-aware transformer.** Dropout of probability 0.5 at the hidden layer of the classifier, combined with learning rates between the orders of magnitude 1e-4 to 1e-5 worked well across different iterations of our datasets and prediction tasks. We always used the same learning rate for a model's fine-tuning as it was pre-trained with. We found that some minor pre-epoch exponential learning rate decay ($\approx 0.9$) lessened the deterioration due to over-fitting after it reached its peak performance but the practical effect of this was marginal on experiments - after just a few epochs, both transformers and value-aware transformers were at their best.

**Preprocessing of data for experiments.** Minimal processing of the data was performed so as not to collapse exploitable signal within the raw data. The visits in the Train set were used to extract per-lab quantile distributions following the procedure outlined in Section 2.4. From these derived distributions, we encoded each lab-event in the Train, Dev. and Test sets as (`[LAB]`, `[VAL]`), with `[VAL]` $\in$ { `[XLOW]`, `[LOW]`, `[MID]`, `[HIGH]`, `[XHIGH]`, `[UNK]` }. Finally, with visits rep-

resented as sequences of token-value pairs, we append a special end-of-sequence token `[EOS]` to mark the 48 hour point.

**Experiment 1** We chose the input sequence length to be 250 for each of the sequence models, because this sequence length choice captures the entirety of 90% of the visits in the Train set (cf. Table 5). For the value-aware transformer we found that setting the loss hyperparameter to $\gamma = 0.9$ was most effective, putting greater weight on the learning of the tokens and counteracting the value token's greater at-random probability of being correctly predicted. Since the number of tokens is $\approx 750$ its baseline probability of correct prediction is approximately 0.0013, while the number of quantilised values is $\approx 10$, hence the baseline chance of correctly predicting a `[VAL]` is orders of magnitude higher, at around 0.1. Learning rates across the order of magnitude 1e-4 to 1e-5 were consistently performant for our data for both transformers and value-aware transformers. We used gradient-clipping and a small amount of dropout in both feedforward and attention layers. We trained each until it hit an early-stopping threshold of 7 epochs without improvement on the Dev. set.

**Experiment 2** To determine a competitive setting for the FFNN baselines we ran a hyperparameter search against the Dev. set. The all-round best-performing hyperparameters from the search were then chosen and 5 separate training runs were performed for each model, checkpointed at the epoch with the best Dev. set loss. For the transformers and value-aware transformers, we fine-tuned the 4 pre-trained models from Experiment 1 (Section 4.2) five times each from scratch, checkpointing them at the epoch with the lowest Dev. set loss. The loss for all models was the same: class-weighted binary cross-entropy. The 5 runs of each competitor were then loaded from their checkpoints and evaluated against Test set, with the mean $\pm$ standard deviation of key metrics reported in Table 3. The Dev. set performance is detailed in Tables 7 & 8. Each model was trained until it hit an early stopping threshold of 3 epochs without improvement on the Dev. set.

**Experiment 3** Exactly the same procedure was followed as for Experiment 2, except that the MSE loss was used for all models. The Test set results are reported in Table 4. The Dev. set performance is detailed in Table 9.

## C   DEV. SET PERFORMANCE FOR MAIN PAPER EXPERIMENTS

Table 7: In-hospital mortality $\leq 3$ days classification performance on the Dev. set

| Model | Data dim. | Mortality $\leq$ 3 days | |
| --- | --- | --- | --- |
| | | Balanced-accuracy | AUROC |
| FFNN raw values (mean) | 1 v | $0.557 \pm 0.049$ | $0.638 \pm 0.024$ |
| FFNN quants (mean, one-hot) | 1 v | $0.556 \pm 0.077$ | $0.784 \pm 0.008$ |
| FFNN quants (mean, integer) | 1 v | $0.604 \pm 0.037$ | $0.732 \pm 0.008$ |
| Transformer (depth 4) | 1 h | $0.698 \pm 0.020$ | $0.800 \pm 0.012$ |
| Transformer (depth 6) | 1 h | $0.694 \pm 0.013$ | $0.797 \pm 0.009$ |
| VA-Transformer (depth 4) | 1.5 | $\mathbf{0.737 \pm 0.049}$ | $\mathbf{0.845 \pm 0.019}$ |
| VA-Transformer (depth 6) | 1.5 | $\mathbf{0.726 \pm 0.020}$ | $\mathbf{0.831 \pm 0.014}$ |

Table 8: In-hospital mortality $> 7$ days classification performance on the Dev. set

| Model | Data dim. | Mortality $>$ 7 days | |
| --- | --- | --- | --- |
| | | Balanced-accuracy | AUROC |
| FFNN raw values (mean) | 1 v | $0.616 \pm 0.041$ | $0.649 \pm 0.055$ |
| FFNN quants (mean, one-hot) | 1 v | $0.696 \pm 0.007$ | $0.784 \pm 0.015$ |
| FFNN quants (mean, integer) | 1 v | $0.691 \pm 0.008$ | $0.779 \pm 0.003$ |
| Transformer (depth 4) | 1 h | $0.685 \pm 0.004$ | $0.773 \pm 0.004$ |
| Transformer (depth 6) | 1 h | $0.687 \pm 0.008$ | $0.772 \pm 0.003$ |
| VA-Transformer (depth 4) | 1.5 | $\mathbf{0.727 \pm 0.005}$ | $\mathbf{0.817 \pm 0.006}$ |
| VA-Transformer (depth 6) | 1.5 | $\mathbf{0.731 \pm 0.007}$ | $\mathbf{0.816 \pm 0.004}$ |

Table 9: Remaining length-of-stay regression performance on the Dev. set

| Model | Data dim. | Remaining LOS at 48hrs | |
| --- | --- | --- | --- |
| | | MSE | $R^2$ |
| FFNN raw values (mean) | 1 v | $158.9 \pm 3.7$ | $0.106 \pm 0.021$ |
| FFNN quants (mean, one-hot) | 1 v | $131.2 \pm 1.0$ | $0.262 \pm 0.006$ |
| FFNN quants (mean, integer) | 1 v | $131.1 \pm 1.3$ | $0.262 \pm 0.007$ |
| Transformer (depth 4) | 1 h | $130.1 \pm 1.2$ | $0.268 \pm 0.007$ |
| Transformer (depth 6) | 1 h | $129.6 \pm 1.0$ | $0.271 \pm 0.006$ |
| VA-Transformer (depth 4) | 1.5 | $\mathbf{124.3 \pm 1.9}$ | $\mathbf{0.300 \pm 0.011}$ |
| VA-Transformer (depth 6) | 1.5 | $\mathbf{123.3 \pm 2.2}$ | $\mathbf{0.306 \pm 0.012}$ |

# D   TEMPORAL EMBEDDINGS

Each of the events in our experiments come with a timestamp. The natural way to utilise this timestamp with a (value-aware) transformer is for the positional embedding to be a function of time. There are now a number of variations on the absolute positional embeddings introduced in Vaswani et al. (2017), with two prominent variations being relative positional embeddings and rotary embeddings (see Wang (2021) for implementations of these, and more). The simplest approach is to use the version of absolute positional embedding detailed in Horn et al. (2020). To add the temporal embedding indicating time $t \in [0, T]$ to a vector $v \in \mathbb{R}^d$, we first the vector $\tau \in \mathbb{R}^d$ with components

$$\tau_{2i} = \sin(t/T^{2i/d})$$
$$\tau_{2i+1} = \cos(t/T^{2i/d})$$

for indices satisfying $0 \le 2i, 2i+1 \le d-1$ and where $T$ is the maximum time relevant for the problem at hand. This vector is then added component-wise to each concatenated token-value embedding, before being fed into the attentional section of the model. While we extracted this information during preprocessing, using it in the experiments would give further advantage to the transformer and value-aware transformers over the non-sequential models, and we wanted this comparison to be as fair as possible.

# E   STRONG-COUPLING EXPERIMENT

For the value-aware transformer we chose to use a loose-coupling method to input the 1.5D data using an embed-concatenate-embed input structure. This scales favourably: with the number of variable tokens $T$ and the number of value tokens $V$, loose-coupling requires a vocabulary of $T + V$ distinct input tokens, while strong-coupling requires a vocabulary of $TV$. For large $T$ and $V$ this is important - especially on smaller datasets consisting of sparse data, where fine quantilisation would result in considerable token sparsity issues. For our experimental set-up with $V = 8$ and $T = 755$, strong-coupling does in fact provided a competitive inductive bias, performing worse than loose-coupling on LOS (Table 11), comparably on Mortality $\le 3$ days (Table 12), but better on Mortality $> 7$ days (Table 13). This could be due to $V$ being small in our experiment, so that loose-coupling does not have the advantage it would were $V$ of a larger size.

Table 10: Pre-training performance: a transformer using the 1.5D data with strong-coupling on Dev. set. $\mathcal{L}_{\texttt{[TOK-VAL]}}$ indicates the cross-entropy loss, and $p_{\texttt{[TOK-VAL]}}$ the corresponding average probability of correct next-token prediction.

| Model | Input form | $\mathcal{L}_{\texttt{[TOK-VAL]}}$ | $p_{\texttt{[TOK-VAL]}}$ |
|---|---|---|---|
| Random | – | – | 0.0002 |
| Transformer (depth 4) | [TOK-VAL] | 2.252 | 0.1052 |
| Transformer (depth 6) | [TOK-VAL] | 2.198 | 0.1110 |

Table 11: Comparison with strong-coupling [TOK-VAL] of 1.5D form: remaining length-of-stay on the Test set

| Model | Data dim. | Input form | Remaining LOS at 48hrs | |
| --- | --- | --- | --- | --- |
| | | | MSE | $R^2$ |
| **Dev. set** | | | | |
| Transformer (depth 4) | 1.5 | [TOK-VAL] | $127.6 \pm 1.2$ | $0.281 \pm 0.007$ |
| Transformer (depth 6) | 1.5 | [TOK-VAL] | $125.7 \pm 1.6$ | $0.292 \pm 0.009$ |
| VA-Transformer (depth 4) | 1.5 | ([TOK], [VAL]) | $124.3 \pm 1.9$ | $0.300 \pm 0.011$ |
| VA-Transformer (depth 6) | 1.5 | ([VAL], [VAL]) | $\mathbf{123.3 \pm 2.2}$ | $\mathbf{0.306 \pm 0.012}$ |
| **Test set** | | | | |
| Transformer (depth 4) | 1.5 | [TOK-VAL] | $128.8 \pm 0.9$ | $0.261 \pm 0.005$ |
| Transformer (depth 6) | 1.5 | [TOK-VAL] | $128.4 \pm 2.8$ | $0.264 \pm 0.016$ |
| VA-Transformer (depth 4) | 1.5 | ([TOK], [VAL]) | $126.9 \pm 1.4$ | $0.273 \pm 0.008$ |
| VA-Transformer (depth 6) | 1.5 | ([VAL], [VAL]) | $\mathbf{125.2 \pm 3.0}$ | $\mathbf{0.282 \pm 0.017}$ |

Table 12: Comparison with strong-coupling [TOK-VAL] of 1.5D form: in-hospital mortality $\leq 3$ days

| Model | Data dim. | Input form | Mortality $\leq$ 3 days | |
| --- | --- | --- | --- | --- |
| | | | Balanced-accuracy | AUROC |
| **Dev. set** | | | | |
| Transformer (depth 4) | 1.5 | [TOK-VAL] | $\mathbf{0.767 \pm 0.019}$ | $0.839 \pm 0.006$ |
| Transformer (depth 6) | 1.5 | [TOK-VAL] | $0.737 \pm 0.049$ | $0.842 \pm 0.015$ |
| VA-Transformer (depth 4) | 1.5 | ([TOK], [VAL]) | $0.737 \pm 0.049$ | $0.845 \pm 0.019$ |
| VA-Transformer (depth 6) | 1.5 | ([TOK], [VAL]) | $0.726 \pm 0.020$ | $0.831 \pm 0.014$ |
| **Test set** | | | | |
| Transformer (depth 4) | 1.5 | [TOK-VAL] | $\mathbf{0.749 \pm 0.041}$ | $0.841 \pm 0.019$ |
| Transformer (depth 6) | 1.5 | [TOK-VAL] | $0.734 \pm 0.030$ | $0.831 \pm 0.016$ |
| VA-Transformer (depth 4) | 1.5 | ([TOK], [VAL]) | $0.725 \pm 0.039$ | $0.825 \pm 0.020$ |
| VA-Transformer (depth 6) | 1.5 | ([TOK], [VAL]) | $0.716 \pm 0.025$ | $0.813 \pm 0.012$ |

Table 13: Comparison with strong-coupling [TOK-VAL] of 1.5D form: in-hospital mortality $> 7$ days

| Model | Data dim. | Input form | Mortality $>$ 7 days | |
| --- | --- | --- | --- | --- |
| | | | Balanced-accuracy | AUROC |
| **Dev. set** | | | | |
| Transformer (depth 4) | 1.5 | [TOK-VAL] | $0.734 \pm 0.006$ | $0.820 \pm 0.004$ |
| Transformer (depth 6) | 1.5 | [TOK-VAL] | $0.729 \pm 0.006$ | $0.819 \pm 0.003$ |
| VA-Transformer (depth 4) | 1.5 | ([TOK], [VAL]) | $0.727 \pm 0.005$ | $0.817 \pm 0.006$ |
| VA-Transformer (depth 6) | 1.5 | ([TOK], [VAL]) | $0.731 \pm 0.007$ | $0.816 \pm 0.004$ |
| **Test set** | | | | |
| Transformer (depth 4) | 1.5 | [TOK-VAL] | $0.723 \pm 0.010$ | $0.816 \pm 0.006$ |
| Transformer (depth 6) | 1.5 | [TOK-VAL] | $0.727 \pm 0.007$ | $0.807 \pm 0.003$ |
| VA-Transformer (depth 4) | 1.5 | ([TOK], [VAL]) | $0.723 \pm 0.016$ | $0.808 \pm 0.003$ |
| VA-Transformer (depth 6) | 1.5 | ([TOK], [VAL]) | $0.724 \pm 0.005$ | $0.810 \pm 0.003$ |

