# OpenReview forum: "Value-aware transformers for 1.5d data"
_ICLR.cc/2022/Conference — ICLR 2022 Submitted_

### Official Review · Reviewer_ph4X · 2021-10-30

**Correctness:** 3
**Technical Novelty And Significance:** 2
**Empirical Novelty And Significance:** 2
**Recommendation:** 6
**Confidence:** 3

**Main Review:**

The paper tackles an important challenge with modeling sequential data in healthcare, sparsity of events-- labs are generally ordered based on the patient context and may be ordered multiple times and it is important to account for the sequential nature of the event as well as the value for each event. The concept that is termed as 1.5-dimensional setting proposes to account for the value when learning the embeddings for the token which are the lab events with their values, an interesting approach for using transformers. However, there are certain concerns with respect to the design choices regarding the architecture as well as the training process such as the decay rate, the dimensions of the hidden layer, $\gamma$. Moreover, the performance that is achieved for say in-hospital mortality is only sub-par with respect to other approaches for predicting in-hospital mortality that are largely neglected in this work. This creates another problem to assess the merits of the proposed method. While there has been considerable work (see 1-5 below), for predicting in-hospital mortality using MIMIC data, comparison with such baselines needs to be performed.

The paper can also be improved to make it better for the reader. Adding why the value-aware representations would be beneficial for downstream tasks in the introduction would help to situate the need for this work. Similarly, adding context about the heterogeneity of the data for patients' individual health can help. The choice for the number of dimensions for the token and value representation needs to be supported. While that for the values can be justified based on the quantization, there are concerns with the generalizability of the quantization process as well. I would encourage the authors to think about the sensitivity of the quantization to the population incorporated in the study. The eICU dataset provides a richer ground to handle these generalizability issues and could be checked.


1. Song, Huan, et al. "Attend and diagnose: Clinical time series analysis using attention models." Thirty-second AAAI conference on artificial intelligence. 2018.
2. Purushotham, Sanjay, et al. "Benchmark of deep learning models on large healthcare mimic datasets." arXiv preprint arXiv:1710.08531 (2017).
3. Che, Zhengping, et al. "Recurrent neural networks for multivariate time series with missing values." Scientific reports 8.1 (2018): 1-12.
4. Jarrett, Daniel, et al. "Clairvoyance: A pipeline toolkit for medical time series." International Conference on Learning Representations. 2020.
5. Li, Ke, et al. "Predicting in-hospital mortality in ICU patients with sepsis using gradient boosting decision tree." Medicine 100.19 (2021).

**Summary Of The Paper:**

The paper proposes an approach for learning representations of spare sequential data, commonly found in healthcare settings. The work challenges issues with sequential data in electronic health records such as sparsity of events, unequal lengths, which also present imputation challenges. A transformer-based approach is presented which incorporates information about the different sequential events (labs) along with their values which is termed as a 1.5-dimensional representation. The transformer is trained to obtain the representation of sequences of events, labs conducted for in-hospital patients, along with the values of the labs which are categorized based on the quantile range obtained from the training data. The representations from the transformer embeddings are used for predicting two outcomes, 1) in-hospital mortality, and 2) length of stay at 48 hours after admission. The value- aware transformer performs marginally better than a value-unaware transformer with respect to $\mathcal{L_{\text{\[TOK\]}}}$. There is also an improvement in the balanced-accuracy and ROC-AUC for in-hospital mortality when compared with feed-froward neural network and value-unaware transformer. Similar performance characteristics are also observed for length-of-stay regression.

**Summary Of The Review:**

In general, the paper suffers from clarity as well as comparison with existing baseline other than transformer-based approaches and feed-forward networks. Assessing the performance of the model on different subpopulations based on demographics would be helpful to identify how the method performs subgroups rather than on average since it is known that there are differences in lab-orderings based on demographic attributes such as gender and self-reported race. This could strengthen the contributions.

---

> ### Author Response · Authors · 2021-11-19
> **[Part 2 / 2]**
>
> > The choice for the number of dimensions for the token and value representation needs to be supported. While that for the values can be justified based on the quantization, there are concerns with the generalizability of the quantization process as well.
>
> Thank you, another reviewer also raised this point: this was an informed, but not an optimised, choice. We chose them so that both the tokens and values were forced to be compressed for their embedding (embedding dimension strictly less than vocabulary size). But we didn't focus on this consideration because of a paper (now referenced) which suggests that over-parameterisation of the token embedding doesn't effect significant degradation in performance.
>
> > I would encourage the authors to think about the sensitivity of the quantization to the population incorporated in the study.
>
> We have clarified the thinking behind this (choosing a quantilisation highlighting the extremities of the physiological measurements) in the revised manuscript. The particular quantilisation we suggested was not intended to be prescriptive - in practise, this would be informed by the particular application. For instance, it may be that regular deciles, normal/abnormal or a more exotic quantilisation could be appropriate for a particular application.
>
> > as well as comparison with existing baseline other than transformer-based approaches and feed-forward networks.
>
> This is something which we do intend to address as soon as we can.
>
> > Assessing the performance of the model on different subpopulations based on demographics would be helpful to identify how the method performs subgroups rather than on average since it is known that there are differences in lab-orderings based on demographic attributes such as gender and self-reported race.
>
> Ethical calibration is exceptionally important for healthcare applications and we wholeheartedly agree that this point is a key consideration in an end-product. For the purpose of this study, we focused entirely on the effective extraction and utilisation of signal from a difficult data regime, and we had to restrict the scope to try and derive a clear message. Your proposal is an important point that we will bear in mind as we develop the model further.
>
> **We hope that our answers and the revisions we have made go some way to addressing your comments so far. Many thanks! We would very much like to hear any more comments or further concerns, so that we can keep working towards improving the manuscript.**

---

> ### Author Response · Authors · 2021-11-19
> **Dear Reviewer ph4X [Part 1 / 2]**
>
> Many thanks for your careful reading of the manuscript, for sharing the references and your comments! We will address each of the main concerns that you raised here, highlighting any changes/clarifications we have made in the revised paper in light of them.
>
> > there are certain concerns with respect to the design choices regarding the architecture as well as the training process such as the decay rate, the dimensions of the hidden layer.
>
> We kept the classifier head specification exactly the same (for each of the FFNNs, transformers and value-aware transformer) so as to make a fair comparison across all models. The input representation(/embedding) was the part that we wanted to investigate the effect of changing. The choice of 100 might not be optimal, but we needed it to be consistent so as to keep the analysis as scientific as we could. The decay of the learning rate was only once per epoch and set to be of low impact (a factor of 0.9) - this was only used on the larger models to help keep them at top performance for slightly longer. Since they typically finetuned for only a couple of epochs before being at their peak, the practical effect of this was marginal in reality. We have clarified this point in a new Appendix B.
>
> > Moreover, the performance that is achieved for say in-hospital mortality is only sub-par with respect to other approaches for predicting in-hospital mortality that are largely neglected in this work. This creates another problem to assess the merits of the proposed method. While there has been considerable work (see 1-5 below), for predicting in-hospital mortality using MIMIC data, comparison with such baselines needs to be performed.
>
> We agree fully with the spirit of your point: there are benchmarks out there - why haven't we used them? We have tried to do a much better job explaining the reasoning behind this choice - we had examined them prior to beginning this study and did benefit from their work. We have devoted a section of the revised Related Work subsection to address this point. Par is difficult to assess in this context (authors in the cited papers agree, even when the cohorts and variables used are largely the same). Our results are not wildly different to those claiming SOTA, in spite of the fact that the mortality problems that we have defined are typically much more imbalanced than those seen elsewhere, and that our models were forbidden from seeing static variables such as demographics, which capture a fair amount of signal by themselves. This is so that a fairer comparison between the different treatments of the sequential data could be made more easily, unconflated with the static data. In short, the point of the paper was to explore how to best utilise the sequence data in a particular data regime, not to optimise for the specified downstream tasks using all data available within MIMIC. Hopefully we have helped make this clearer with a complete rework of the Introduction so as clarify the motivation for the study. Thank you for raising this point!
>
> > The paper can also be improved to make it better for the reader. Adding why the value-aware representations would be beneficial for downstream tasks in the introduction would help to situate the need for this work. Similarly, adding context about the heterogeneity of the data for patients' individual health can help.
> >  In general, the paper suffers from clarity
>
> Thank you for raising this! We have made substantial modifications to the manuscript, especially in the introduction, highlighting why the data regime we are addressing is challenging, motivated by statistics from the dataset as well as numbers of imputations required for different data representations. We have reorganised the paper so that our contributions are clearer, and done better justice to both the works that influenced this one and the important healthcare benchmarks which we examined prior to beginning this work. We hope we have done a better job of motivating the reasoning behind this study through addressing your concerns.

---

> > ### Comment · Reviewer_ph4X · 2021-11-29
> > **Thank you for your response**
> >
> > I thank the authors for the clarifications. I have carefully read the authors' responses as well as other comments. The authors have addressed most of my concerns. I have increased the score to 6 and believe that the problem is interesting and the work paves an important direction.

---

> > > ### Author Response · Authors · 2021-11-30
> > > **Dear Reviewer ph4X**
> > >
> > > Thank you for your careful reading of the paper, and for your support in improving it. We will try and address any outstanding concerns as we move forward with the project!

---

### Official Review · Reviewer_qFRi · 2021-11-01

**Correctness:** 3
**Technical Novelty And Significance:** 2
**Empirical Novelty And Significance:** 2
**Recommendation:** 3
**Confidence:** 5

**Main Review:**

Positives:
1. The paper focuses on the task of modeling sparse and multivariate sequential data which is an important and practical problem.
2. The paper is easy to follow.

Concerns:
1. The authors are missing several important related works and comparisons with recent and SOTA methods for sparse and multivariate sequential data [1, 2, 3].
2. The main contribution of the paper is not clear.
3. The authors focus on the data representation for the sparse and multivariate sequential data but fail to compare it with some of the similar data representations introduced in [3] and [4].
4. The proposed framework has marginal novelty in terms of the model architecture as it is a fairly standard transformer architecture.
5. The authors should provide experiments to show if the design principles hold in practice, in particular, provide a comparison with strong coupling.
6. The authors should perform experiments with more datasets to show the effectiveness of the model.

Additional Comments:
1. How did the authors choose the dimension of the token and value embedding?
2. Why did the authors choose to not consider the temporal information such as the time of the lab events which is fairly common to consider in several works in this domain?
3. What happens if multiple lab events are observed at the same time? How is the order decided in that case?
4. Could the authors clarify more about the difference between CLS and EOS token and why it has to be added in the end? Based on my understanding of transformer architectures it doesn't really matter where the token is added.
5. Is the padding used because of the variable length of sequence from different examples?


References:
1. Zhengping Che, Sanjay Purushotham, Kyunghyun Cho, David Sontag, and Yan Liu. Recurrent neural
networks for multivariate time series with missing values. Scientific Reports, 8(1):6085, 2018.
2. S. N. Shukla and B. Marlin. Multi-time attention networks for irregularly sampled time series. In International Conference on Learning Representations, 2021.
3. Max Horn, Michael Moor, Christian Bock, Bastian Rieck, and Karsten Borgwardt. Set functions for
time series. In Proceedings of the 25th International Conference on Machine Learning, 2020.
4. S. N. Shukla and B. M. Marlin. A Survey on Principles, Models, and Methods for Learning from Irregularly Sampled Time Series. CoRR, abs/2012.00168, 2020.


**Summary Of The Paper:**

This work proposes an approach for modeling sparse and multivariate sequential data which are commonly found in the healthcare domain. The paper presents and compares different data representations and introduces a 1.5d representation and outlines principles for modeling this type of data. The paper presents the value-aware transformer, a transformer decoder-only architecture for the 1.5d data representation. Based on the experiments on the MIMIC-III dataset, the proposed model and the data representation achieve better performance than several other standard data representations.

**Summary Of The Review:**

The contributions of the paper are not clear. The paper is also missing important related works and comparisons with recent approaches in this domain. The paper has only marginal novelty both in terms of the data representation and the proposed approach.

---

> ### Author Response · Authors · 2021-11-19
> **[Part 2 / 2]**
>
> > Why did the authors choose to not consider the temporal information such as the time of the lab events which is fairly common to consider in several works in this domain?
>
> While we extracted this information during pre-processing, using it in the experiments would give further advantage to the transformer and value-aware transformers over the non-sequential models, and we wanted this comparison to be as fair as possible. This information is often used though, as you say. We have added a section detailing a natural approach to using this information in Appendix D using temporal embeddings.
>
> > What happens if multiple lab events are observed at the same time? How is the order decided in that case?
>
> We have added a paragraph describing this is Section 2.4. For our application, if labs are approximately coincident, then they their order should not affect subsequent investigation much. To help the model learn these coincident patterns we used a consistent ordering (based on the empirical frequency of the tokens). We don't expect the specific ordering we used to be important, although that is something further to investigate! This can be further addressed using the temporal embeddings of Appendix D.
>
> > Could the authors clarify more about the difference between CLS and EOS token and why it has to be added in the end? Based on my understanding of transformer architectures it doesn't really matter where the token is added.
>
> Happy to clarify: there are a number of common styles of transformer: encoder-only (e.g. BERT), encoder-decoder (commonly used in neural machine translation), decoder-only (e.g. GPT). Each has its own inductive bias and uses its own training methods - the decoder-only style we used has "future-masking" as its inductive bias. This stops it from looking as subsequent input tokens (and their embeddings) while generating the current output token - only information from previous tokens can be used in predicting the next one.
> If we were to prepend a token to each input sequence for this style of transformer, then every first position output embedding would be identical, because the model has seen no other information at that stage. During finetuning, a classifier using that output embedding for supervised learning would see identical input information, together with the data instance's target label, so that it wouldn't be able to learn anything interesting about the task. We have highlighted this in the revised manuscript - please do ask more in case further clarification is helpful.
>
> > Is the padding used because of the variable length of sequence from different examples?
> Yes, we have touched on this in section 3.2.1 the and included this padding in our imputation comparisons in Table 6, although it is worth highlighting that the pad tokens are not used to calculate the model losses (so it is not imputation in a true sense).
>
> **We hope that our answers, the substantial changes to the manuscript, and the further experiment has helped address some of your concerns. We would very much like to hear any more comments if you have time so that we can improve the manuscript further. Many thanks!**

---

> ### Author Response · Authors · 2021-11-19
> **Dear Reviewer qFRi [Part 1 / 2]**
>
> Many thanks for highlighting the contemporary work on irregular time series, and for your clear and structured review. The antecedents of our work were focused on the tokenised sequence learning of EHRs, and it is refreshing to see related work coming in from an irregular  time-series angle. We have made substantial revisions in light of your concerns and comments.
>
> In particular, we didn't make the properties of the data regime we are considering clear enough - we have addressed this throughout the paper and hope it is now clearer how extremely sparse the data is in comparison with some of the time series analyses referenced. It is especially the highly-multivariate nature, together with the sparsity, which makes the data regime we are studying in this paper challenging.
>
> > The authors are missing several important related works and comparisons with recent and SOTA methods for sparse and multivariate sequential data [1, 2, 3].
>
> Thank you, we have examined and referenced these papers, considerably expanded the Related Work (Section 1.2) highlighting their contributions.
>
> > The main contribution of the paper is not clear.
>
> Thank you, we have now made this clearer, in a new "Our Contributions" subsection on the first page.
>
> > The authors focus on the data representation for the sparse and multivariate sequential data but fail to compare it with some of the similar data representations introduced in [3] and [4].
>
> We have highlighted these works with their own dedicated subsection of the Related Work - excellent references, thank you! The representation that we are proposing is quite different from each of the three listed in [4]: it is perhaps closest to the set-based, but with two important differences: one that the sequential structure is explicitly maintained, and second that we map raw values to a unified (tokenised) scale via the quantilisation.
>
> > The proposed framework has marginal novelty in terms of the model architecture as it is a fairly standard transformer architecture.
>
> Our aim here was to propose modifications that are largely orthogonal to the incredibly active work on transformer architectures. We intentionally used a relatively basic transformer, and then showed how to modify it. There is a huge and ever-expanding body of work on the internals and efficiency of the transformer architecture - our work compliments that by modifying the input and output to work on a different data form. We are not aware of work that has done what we have proposed.
>
> > The authors should provide experiments to show if the design principles hold in practice, in particular, provide a comparison with strong coupling.
>
> Thank you for the suggestion, we have conducted this experiment and commented on it in Appendix E. The upshot is that it is also competitive within our experimental set-up. We have also commented on representation efficiency with respect to the sizes of the [TOK] (T) and [VAL] (V) vocabularies. The original motivation for loose-coupling was the representation efficiency: loose-coupling requiring a vocab of size V + T while strong-coupling requires a vocab of VT, which will cause considerable token sparsity issues for as V grows larger. Our dataset pre-processing and experimental set-up wasn't chosen to illustrate this point, but it is a fair one to raise, thank you.
>
> > The authors should perform experiments with more datasets to show the effectiveness of the model.
>
> This would strengthen the empirical evidence and we will try and address this as soon as possible. The particular dataset we used is a perfect example of the data regime we are trying to work with, which was why we chose it. We have shown that we have a competitive advantage over a plain transformer here - and transformers have been highly competitive for related problems such as Physionet 2012, albeit with curated variables (e.g. empirical study in the SeFT paper).
>
> > How did the authors choose the dimension of the token and value embedding?
>
> Thank you for raising: this was an informed, but not an optimised, choice. We chose them so that both the tokens and values were forced to be compressed for their embedding (embedding dimension strictly less than vocabulary size). But we didn't optimise on this because of a paper [Yin, Zi and Shen, Yuanyuan: On the dimensionality of word embedding, 2018] which suggests that over-parameterisation of the token embedding doesn't effect significant degradation in performance for many common word-embedding algorithms.

---

> > ### Comment · Reviewer_qFRi · 2021-11-30
> > **Response follow up**
> >
> > I would like to thank the authors for responding to my concerns. I have decided to keep my score because my concerns haven't been fully addressed.
> >
> > I think all the three contributions mentioned in the revised paper are marginal and unconvincing. First, the data encoding is very similar to the data representation of Horn et al [3] and the set-based representation introduced in [4]. For transformer models used in the paper, sequential nature is only maintained through the positional embedding while in [3] the sequential nature is maintained by a time encoding of the observed time points. Furthermore, Horn et al also consider a Transformer based approach where they replace the positional encoding with a time embedding. These approaches has been discussed in detail in Section 8 of the survey paper [4].  Second contribution is not clear. The loose coupling or concatenation of the value and token embedding is pretty standard and has been studied for time series in one of the baselines of [3] and Xu et al. (2019). Furthermore, based on additional experiments in Appendix E it is not clear if the loose coupling is optimal in case of quantized values. If the authors refer to the quantization as the contribution here then that would require further experimental evidence and reasoning. I'm also not convinced on the third contribution regarding the empirical evidence as the comparison with SOTA methods (pointed out in the main review) are still missing and its hard to judge the performance based on a single dataset.
> >
> > References:
> >
> > [Xu et al.] Da Xu, Chuanwei Ruan, Evren Korpeoglu, Sushant Kumar, and Kannan Achan. Self attention with functional time representation learning. In Advances in Neural Information Processing Systems, pages 15915–15925. 2019.
> >
> > *Other reference numbers correspond to the reference list from the main review.

---

> > > ### Author Response · Authors · 2021-11-30
> > > **[Follow-up part 2 / 2]**
> > >
> > > >  The loose coupling or concatenation of the value and token embedding is pretty standard and has been studied for time series in one of the baselines of [3] and Xu et al. (2019).
> > >
> > > This statement is incorrect - at least based on all of the papers we have read and the papers that the reviewer has presented. We are yet to see a paper that has taken the approach that we have outlined.
> > >
> > > The baseline in [3] does not use loose-coupling, it uses a raw vector multivariate representation with imputation of 0 for missing values (see [3] A.2 & A.3), as mentioned in more detail in the comment above.
> > >
> > > The new reference the reviewer has highlighted, Xu et al. (2019), is a study concerning functional time-representation learning. This is neither a transformer model nor does it utilise the loose-coupling of variable-value token pairs. They study the effect of concatenating different styles of functional time-representations to event embeddings as a means of encoding position(/temporality). It is difficult to draw a parallel between this reference and the work we have presented, since both the problem being considered, and the approach taken, are different.
> > >
> > > > Furthermore, based on additional experiments in Appendix E it is not clear if the loose coupling is optimal in case of quantized values.
> > >
> > > The additional experiments in Appendix E that we undertook at your suggestion show that the loose-coupling method beats the baselines on all tasks, and out-performs the strong-coupling by a healthy margin on at least one of the tasks, performing competitively on others.
> > >
> > > > I'm also not convinced on the third contribution regarding the empirical evidence as the comparison with SOTA methods (pointed out in the main review) are still missing and its hard to judge the performance based on a single dataset.
> > >
> > > Following your review, we took special effort to emphasise through computations, dataset statistics and application-relevant considerations, why our data regime is different to those motivating the time-series literature you highlighted. We intentionally didn't use "SOTA" in the third bullet of the 'Our contributions' section because state-of-the-art is uninformative when modelling in a new context. We  intend to conduct further experiments as we continue to explore modelling in the sparse sequential highly-multivariate regime.
> > >
> > > &nbsp;
> > >
> > >
> > > We hope these clarifications and the additional detailed references help to clear up some of the apparent misunderstanding. If we can help by providing further explanation focused on any of the points above or elsewhere, we would be glad to do so.
> > >
> > > To summarise, we have considered the works that you have referenced carefully and improved the original manuscript during the rebuttal stage in light of your concerns. We clarified our contributions, referenced and discussed the relevant time-series literature, and presented the results of further empirical experiments at your suggestion. We will continue to work towards strengthening the work, and we thank the reviewer for their engagement.

---

> > > ### Author Response · Authors · 2021-11-30
> > > **Dear Reviewer qFRi [Follow-up Part 1 / 2]**
> > >
> > > Thank you for taking the time to read our submission, the revision, and for your further comments. Since some of these comments suggest a misunderstanding of our manuscript and of the references, we have provided detailed clarifications below:
> > >
> > > &nbsp;
> > >
> > > > First, the data encoding is very similar to the data representation of Horn et al [3] and the set-based representation introduced in [4].
> > >
> > > In contrast with [3, 4]'s set-based representation (an unordered collection of events), we are retaining and exploiting the sequential structure, without requiring the substantial imputation needed for the multivariate vector representation detailed in [4]. We had addressed this succinctly in the revised Related Work section (see 'Irregular time series'). Another key difference is that [3] represents values in their raw form, while we transform them on a per-variable basis and then tokenise. This means that the values can be modelled using a coherent loss function at the pre-training stage.
> > >
> > > > For transformer models used in the paper, sequential nature is only maintained through the positional embedding while in [3] the sequential nature is maintained by a time encoding of the observed time points. Furthermore, Horn et al also consider a Transformer based approach where they replace the positional encoding with a time embedding. These approaches has been discussed in detail in Section 8 of the survey paper [4].
> > >
> > > We have read the survey paper [4] in detail. A "transformer based approach" can mean many things and since there was not sufficient detail to go on in [4] Section 8, we examined Horn et al. [3], its appendices (A.2 & A.3) and the source code in the [paper's repo](https://github.com/BorgwardtLab/Set_Functions_for_Time_Series).
> > >
> > > To clarify, they use a standard transformer (encoder-decoder à la Vaswani et al. (2017)) and a raw multivariate vector form as input (with imputation of 0 if values are missing), but adapt the positional embedding to use timestamps (i.e. a temporal embedding). As described in our revised manuscript (Table 6), to use this modelling approach in the data regime we are studying would require 98% imputation on average.
> > >
> > > Their approach is different in method and style to what we have presented here, where the variable being sampled and its value have been separately tokenised and separately attributed a learnable embedding, before pairwise concatenation and positional (/temporal) embedding. Additionally, after the attention layers we use a hierarchical prediction structure for the pre-training stage, amongst other differences pertaining to the fine-tuning stage.
> > >
> > > We have addressed your concern as to how timestamps could be utilised through temporal embeddings in Appendix D. To employ this in the experiments would not have added much novelty (it has been experimented with in [3] and involves very little change to the standard positional embedding). More importantly for the purpose of this study, it would have put the baseline 1D vertical representation methods at further disadvantage, since they collapse information across the temporal dimension. We wanted to make a fair comparison.

---

### Official Review · Reviewer_KQ5r · 2021-11-02

**Correctness:** 3
**Technical Novelty And Significance:** 3
**Empirical Novelty And Significance:** 3
**Recommendation:** 6
**Confidence:** 4

**Main Review:**

> Good motivation: The motivation of solving the representing sparse sequential highly-multivariate data is novel.

> Authors proposed the good idea in overall; however there are few heuristics involved: I agree that proposing the 1.5D representation and value-aware transformer are novel and quite good idea. However, when proposing the 1.5D representation, I could not find the mathematical grounds for few factors: For example, there is neither clear reason nor ablative studies for selecting 5 degreed values (ie. XLow to XHIGH).

> Good performance while few ablative studies are missing: I could find that 1D representation have been explicitly experimented during the experiments; however I could not find the baseline that represents signals in the form of multivariate series data and applies corresponding models. I think this baseline is required as the evidence for authors' insist that the sparse sequential highly-multivariate data could be better represented in 1.5D compared to multivariate representation.


**Summary Of The Paper:**

Authors proposed the method that could represent the sparse sequential highly-multivariate data, which could be trivially represented as neither 1D nor dense multivariate series data. Authors proposed the 1.5D representation which is composed of token-value pair and proposed the value-aware transformer that is able to use the representation. Experimental results on in-patient laboratory data showed the effectiveness of their data representation and algorithms (ie. value-aware transformer).

**Summary Of The Review:**

Due to the reasons I stated in the main review, I am currently in the borderline accept; however effectively rebutting my comments will make me raise my voting towards higher score.

---

> ### Author Response · Authors · 2021-11-19
> **Dear Reviewer KQ5r**
>
> Many thanks for your support in improving the paper! We have made considerable changes and have tried to address your concerns thoroughly.
>
> > I could not find the mathematical grounds for few factors: For example, there is neither clear reason nor ablative studies for selecting 5 degreed values (ie. XLow to XHIGH).
>
> We expect that the optimal quantilisation will depend greatly upon the application. For our application, we chose them so as to emphasise the extremities of each variable's value distributions - this was natural choice since many of the variables are physiological measurements. We did not intend to suggest that this is the only choice or is prescriptive, and have clarified this in the revision. Depending on the problem a binary indicator or a quantilisation based on deciles might be the most appropriate. Interestingly, we found that the 1D vertical collapse FFNN classifiers performed better on the downstream tasks using our quantilisation than using the raw values of the variables, lending some credence to our particular choice for this application. We have added a remark on this in the revised paper (Section 4.3).
>
> > however I could not find the baseline that represents signals in the form of multivariate series data and applies corresponding models. I think this baseline is required as the evidence for authors' insist that the sparse sequential highly-multivariate data could be better represented in 1.5D compared to multivariate representation.
>
> The purpose of this study was to develop a method which didn't require alarming amounts of imputation for the data regime under consideration. We didn't make this as clear as we would have liked in the original submission - thank you for raising this point! We have made it more precise what we mean by the data regime we are addressing, especially in the Introduction. Our main argument against a full multivariate sequence representation is the amount of imputation required for using these forms for our data regime, which is prohibitively large. We have included a table for the MIMIC-III lab-events data (Table 6), which highlights the severity of this issue, showing that on average, over 98% of the data would need to be imputed - even for the most efficient of these multivariate sequence forms, with a vast proportion of variables with no entries at all for any given series. We have also expanded Section 2.3 with a more precise discussion of the dangers of imputation in healthcare data together with references.
>
> **Many thanks for your review - if you have any further comments on the revised manuscript in the remaining days we would be very glad to hear from you, so as to refine it further!**

---

> > ### Comment · Reviewer_KQ5r · 2021-11-29
> > **Thank you for the response.**
> >
> > I originally offered borderline accept for this draft, and I still think that this paper is worth accepted; however some of heuristics made me not going towards the higher score. I think that authors have to show the ablative experiments on their heuristic design choices; if they were not able to verify their optimality.

---

> > > ### Author Response · Authors · 2021-11-30
> > > **Dear Reviewer KQ5r**
> > >
> > > Thank you for your support of the paper! We're sorry that we didn't manage to strengthen your support with the additional discussion and statistics regarding imputation, and the problem-specific motivation behind the quantilisation choices. As we move forward with the project we will conduct further empirical experiments to explore this.

---

### Author Response · Authors · 2021-11-19
**Dear Reviewers**

Many thanks to each of you for a careful read of our manuscript and for your suggestions for improvement: it is excellent that we have benefitted from your multidisciplinary expertise, both in time-series and healthcare domains. We have made substantial modifications to the manuscript, provided extra information and run new experiments in order to address your concerns and comments. We will provide personalised replies to each of your reviews very soon, highlighting these changes. We will do our best to address any further comments or suggestions that you have in the time remaining!

---

### Decision · Program_Chairs · 2022-01-20

**Decision:**

Reject

**Comment:**

The paper propose a value-aware transformer for sparse multivariate time series data. While to approach is well motivated and the problem well-motivated from a clinical viewpoint, the comparison with related work brought up by reviewer qFRi and  reviewer ph4X would really make it clear where this paper stands. The authors attempt to diffuse this issue in their replies, but empirical comparisons in the paper would guide practitioners more. This is especially important as the paper is motivated by a real-world problem.